# Synthesis and Antimicrobial Evaluation of Some New Organic Tellurium Compounds Based on Pyrazole Derivatives

**DOI:** 10.3390/molecules25153439

**Published:** 2020-07-29

**Authors:** Asmaa B. Sabti, Adil A. Al-Fregi, Majeed Y. Yousif

**Affiliations:** 1Department of Chemistry, College of Science, University of Misan, Basrah 62001, Iraq; asmaabadr86.ab@gmail.com; 2Department of Chemistry, College of Science, University of Basrah, Basrah 61004, Iraq; majeedyousef@gmail.com

**Keywords:** pyrazoles, organotellurium, tellurium tetrabromide, transmetalltion reaction, antimicrobial activities

## Abstract

A novel series of organic tellurium compounds based on pyrazole derivatives with a general formula of ArTeBr_3_ and Ar_2_TeBr_2_ [Ar = 2-(3-(4-substituted phenyl) -5-(2-chlorophenyl)-1*H*-pyrazol-1-yl)-3,5-dinitrophenyl] were obtained by the refluxing of corresponding aryl mercuric chlorides with TeBr_4_ in two different mole ratio of 1:1 and 2:1, respectively, in free-moisture dioxane solvent under an argon atmosphere. Compounds of ArTeBr_3_ and Ar_2_TeBr_2_ were reduced by the action of ethanolic solution of hydrazine hydrate obtained Ar_2_Te_2_ and Ar_2_Te, respectively. Reaction of Ar_2_Te_2_ with excess thionyl chloride or iodine gave the corresponding trihalides ArTeCl_3_ and ArTeI_3_, respectively while the reaction of Ar_2_Te with thionyl chloride or iodine gave the corresponding Ar_2_TeCl_3_ and Ar_2_TeI_3_, respectively. The structures were elucidated according to their elemental analysis of carbon, hydrogen and nitrogen (CHN) and some of the spectroscopic techniques such as infrared IR and nuclear magnetic resonance for ^1^H and ^13^C. The antimicrobial activity for all the synthetic compounds were assayed against both Gram-negative and Gram-positive bacteria by using the agar diffusion method. The tellurated pyrazole derivatives showed a good degree against bacteria growth. In some cases, the antimicrobial activities of the synthetic compounds were better than amoxicillin.

## 1. Introduction

Pyrazoles are one of the important five-membered heterocycles compounds that contain three carbons and two adjacent nitrogen atoms and are considered the most famous class among the azole compounds. In fact, different synthesis procedures have been carried out over the decades [1,2]. Generally, the compounds that contain the pyrazole moiety possess many different applications in several fields such as in the fields of pharmaceutical drugs and technology. The literature confirms that pyrazole derivatives are used as antimicrobial, antifungal, antitumor, anti-inflammatory, and antioxidant compounds [3]. A common method for the synthesis of the substituted pyrazoles includes the cyclization reaction of α,β-unsaturated ketones (chalcones) with hydrazines [4,5].

To the best of our knowledge, through a literature search of SciFinder Scholar and Web of Knowledge, there have been few reported studies related to the synthesis of organotellurium compounds derived from pyrazoles [6,7]. Furthermore, due to the high significant antimicrobial activity of both pyrazoles [2,4] and organic tellurium compounds [8,9,10], in addition, to our interest in the field of organic synthesis for tellurium compounds by the transmetalation reactions between aryl mercury chlorides with tellurium tetrabromide [11,12], in the present work, we synthesized some organic tellurium compounds based on pyrazole and evaluated their microbial activity against four different bacteria.

## 2. Results and Discussion

First, substituted chalcones i-iii and 2-hydrazinyl-3,5-dinitrophenylmercury chloride were prepared, starting according to the procedure in the literature [13]. Organomercuric chlorides containing pyrazole moiety 1–3 were obtained from the corresponding substituted chalcones and 2-hydrazinyl-3,5-dinitrophenylmercury chloride (Scheme 1).

Aryltelluium tribromides **4**–**6** are generated from the reaction of tellurium tetrabromide with **1**, **2,** and **3**, respectively, in a 1:1 mole ratio as orange, reddish orange, and yellowish orange solids in moderate yields. In a similar manner, good yields of diaryl tellurium dibromides **7**, **8**, and **9** were resulted by the reaction of tellurium tetrabromide with **1**, **2**, or **3**, respectively, in a 1:2 mole ratio as orange—yellowish orange solids. Reduction of compounds **4**, **5**, and **6** by hydrazine hydrate gave the corresponding ditellurides **10**, **11**, and **12**, respectively, in good yields, while the reduction of **7**, **8**, or **9** gave the corresponding tellurides **13**, **14**, and **15**, respectively, as dark red solids in low to moderate yields. The preparative methods of all the new synthesized compounds **4**–**15** are illustrated in Scheme 2.

Reaction of diaryl ditellurides **10**–**12** with an excess of thionyl chloride or iodine solutions gave the corresponding aryl tellurium trichlorides ArTeCl_3_
**16**–**18**, and triiodides ArTeI_3_
**19**–**21**, respectively (Scheme 3). Reaction of diaryl telluride Ar_2_Te **13**–**15** with thionyl chloride and iodine solutions gave the corresponding diaryl tellurium dichlorides Ar_2_TeCl_2_
**22**–**24** and diiodides Ar_2_TeI_2_
**25**–**27**, respectively (Scheme 4).

All the synthetic compounds were followed by thin-layer chromatography (TLC) since the reaction of chalcones with aryl hydrazines or reaction of tellurium tetrabromide with aryl mercuric chlorides are critical reactions, so we may expect the possibility of two different compounds (substituted pyrazoles and pyrazolines). However, after purification, no indications of undesired compounds were found upon checking the elemental analysis CHN and ^1^H- and ^13^C-NMR spectra. Generally, the prepared compounds **1**–**27** are colored, solid, stable in air, and soluble in DMF and DMSO.

The structures of recrystallized **1**–**27** were identified by means of spectroscopic techniques and elemental analysis. For the elemental analysis of carbon, hydrogen, and nitrogen of compounds **1**–**27**, both the theoretical (calculated) and practical (found) are approximately identical values. This indicates the validity of the suggested structures.

The FTIR spectra of all prepared compounds **1**–**27** showed important vibrational bands in certain regions, especially in the fingerprint (these spectra are available online as Appendix A). The FTIR spectra shows that the disappearance of the stretching band belonged to C=O and C=C bonds of chalcones in the region between the 1650–1662 cm^−1^ and 1597–1604 cm^−1^ regions, respectively, and stretching of N–H bond of hydrazine in the 3200–3500 cm^−1^ region [4,13]. On the other hand, the spectra of pyrazole derivatives showed the stretching bands for C=N at the 1597–1631 and C=C at 1400–1531 cm^−1^ regions [4].

^1^H- and ^13^C- NMR spectra of **1**–**27** were carried out in DMSO-*d*_6_ solutions (these spectra are available online as Appendix A). These compounds appeared for the expected characteristic signals of all protons in these compounds, which are in good agreement with their suggested molecular formula (Figure 1).

A singlet signal equivalent to one or two protons at δ = 5.80–5.96 ppm was characteristic for H4 (and H4′) protons. This signal can be considered as evidence of the prepared pyrazole ring [14,15]. Multiplet signals at 7.45–8.23 ppm can be assigned to protons of the aromatic rings. For compounds **2**, **5**, **8**, **11**, **14**, **17**, **20**, **23**, and **26**, the protons of methyl groups that attached on C21(and C21′) appeared as a singlet signal at δ 2.20–2.22 ppm. For compounds **3**, **6**, **9**, **12, 15**, **18**, **21**, **24**, and **27**, the spectra showed that the singlet signal in the region δ = 3.86–3.89 ppm can be characterized for the OCH_3_ group at C21 (and C21′) [16].

The signals for the carbon atoms of pyrazole rings can be identified via their corresponding ^13^C NMR spectra, which were in excellent agreement with previous papers [14,17]. These signals can be considered as more evidence of the correctness of the suggested structures (Table 1). The ^13^C NMR spectra of **1**–**27** revealed a signal at the range 139.50–139.88 ppm due to C3, while the C5 atoms showed a signal at the range of 149.60–149.80 ppm. The ^13^C NMR spectra of **1**–**27** appeared as a signal at the region of 103.18–104.66 ppm, which can be assigned to C4. Surprisingly, we found the chemical shift of the C4 atom signal effect from the 3-aryl group, which in turn effects the resonance contributions of the 3-aryl group and the chemical shift difference at C4 of the pyrazole ring. We conclude from the above that the 3-aryl group may be located at the same plane of the pyrazole ring, increasing the resonance contribution of π-bonds. The ^13^CNMR spectra of compounds **4**–**27** showed that the signal in the 114.90–114.06 ppm region can be identified as carbon atoms that bear the tellurium-carbon bond *Te-C*. Comparatively, this signal possessed a low chemical shift (high field), which may be due to the polarity of the tellurium–carbon bond. In general, the aromatic carbon signals appear between 124.00–163.43 ppm. The ^13^C NMR spectra of **2**, **5**, **8**, **11**, **14**, **17**, **20**, **23**, and **26** showed that the signal within the region 21.59–21.82 ppm was due to the carbon of the methyl group. Compounds **3**, **6**, **9**, **12**, **15**, **18**, **21**, **24**, and **27** that appeared as a signal in the high field at the range 55.58–56.60 ppm was attributed to the methoxy group.

The ultra violet-visible for compounds **1–27** showed an absorption peak in the region of 340–500 nm due to π→π* of aromatic rings [18]. Upon investigation of the UV absorptions of the synthetic compounds, it was clearly found that electron-withdrawing Br on the phenyl ring attached to pyrazole caused a change in the maximum absorption (λ_max_) values to longer wavelengths (red shift) while electron-donating groups (Me and OMe) moved to shorter wavelengths (blue shift).

It was proven that the molar conductance of organic mercuric compounds **1**–**3**, aryl tellurium trihalides **4**–**6** and **16**–**21**, and the dihalides **7**–**9** and **22**–**27** behaved as for 1:1 electrolytes, which are in good agreement with the previous works in DMSO [11,12,19,20]. These observations may be due to the ionic character of Hg–Cl and one of the Te–halide bonds in these compounds.

The antimicrobial activity for all pyrazole derivatives **1**–**27** were assayed against two different types of organisms. The first type was Gram-positive bacteria represented by *Staphylococcus aureus* ATCC25923 and *Candida albicans* ATCC2091 and the second was Gram-negative bacteria represented by *Escherichia coli* ATCC25922 and *Pseudomonas aeruginosa* ATCC9027. Amoxicillin (10 µg/disc) in DMSO solvent was used as a standard drug by using the agar well diffusion method (Table 2). It can be concluded that all of the compounds (except ArTeX_3_; X = Cl, Br, I) showed potent growth inhibition against both Gram-negative and Gram-positive bacteria in different degrees. Rank of the antimicrobial activity of pyrazole derivatives follow the sequence:Ar_2_Te **13**–**15** > Ar_2_TeX_2_**7**-**9** and **22**–**27** > ArHgCl **1**–**3** ≥ Ar_2_Te_2_**10**–**12** >ArTeX_3_**4**–**6** and **16**–**21**

In general, the telluride compounds **13**, **14**, and **15** were more potently active against the bacteria than the other compounds and the control (amoxicillin). This observation may be attributed to the lipophilic properties of the tellurides, which facilitates digestion of the bacteria cellular membrane, or may perhaps be due to their ability to form hydrogen bonds inside the bacteria cell with some active functional groups.

The values of minimum inhibitory concentration (MIC) for pyrazole derivatives that possessed inhibition zones larger than 10 were tested by the agar well diffusion method as shown in Table 3. The rank of activity was observed as follows: Br > OMe > Me.

## 3. Experimental

### 3.1. Instrumentation

CHN analysis was conducted at the University of Al al-Bayt, Al-Mafraq, Jordan by using a Euro vector EA 3000A elemental analysis (Rome, Italy). FTIR spectra for all synthetic compounds were performed by using a FTIR spectrophotometer Shimadzu model 8400S (Tokyo, Japan) as KBr disk in the range of 4000–400 cm^−1^ at the University of Basrah. ^1^H NMR and ^13^C NMR spectra were measured by using an Ainova (500 MHz) in DMSO-*d*_6_ solution and tetramethyl silane as the internal standard at Tehran University, Tehran, Islamic Republic of Iran. Ultraviolet–Visible spectra for all synthetic compounds were measured at Basrah University, Basrah, Iraqby using Scan 80D (London, England) in the region 200–800 nm by using a chloroform solution 1 × 10^−4^ M and 1 cm^3^ pathway quartz cells. Measurements of molar conductance were performed for all compounds in DMSO solutions of 1 × 10^−3^ M at room temperature by using a Konduktoskop model 365B conductivity bridge. Melting points were made by using a Gallenkamp melting point apparatus (London, England).

### 3.2. Synthesis

#### 3.2.1. General Method for the Preparation of Aryl Mercury(II) Chlorides

2-(3-(4-Substitutedphenyl)-5-(2-chlorophenyl)-1H-pyrazol-1-yl)-3,5-dinitrophenyl)mercury(II) chloride **1**–**3**

A mixture of compound 2-hydrazinyl-3,5-dinitrophenylmercury chloride (3 mmol) in 25 mL of acetic acid and (4 mmol) of chalcones: 3-(2-chlorophenyl)-1-(4-bromophenyl)-prop-2-en-1-one, 3-(2-chlorophenyl)-1-(4-methylphenyl)prop-2-en-1-one and 3-(2-chlorophenyl)-1-(4-methoxy phenyl)prop-2-en-1-one), respectively, was refluxed for 5 h. Then, a catalytic amount of HCl (6–8 drops) was added and the mixture was refluxed for 1 h. After cooling, 50 mL of ice water was added to obtain a yellowish brown solid. The resulting precipitate was filtered, washed several times with water, and recrystallized (twice) from ethanol to obtain yellow solid in 65–77% yields.

2-(3-(4-Bromophenyl)-5-(2-chlorophenyl)-1H-pyrazol-1-yl)-3,5-dinitrophenyl) mercury (II) chloride (**1**)

Light-yellow crystalline solid; Yield: 77%; M.p.: 193–195 °C; Rf = 0.55 (ethyl acetate-n-hexane); Molar conductance (Λ_m_, ohm^−1^ cm^−1^ mol^−1^): 35; FTIR (KBr) cm^−1^: FTIR (KBr) cm^−1^: 3069 w, 1600 s, 1510 s, 1465 s, 1438 s, 1390 s, 1275 m, 1211 s, 1175 m, 1107 m, 1065 m, 1107 m, 1028, 976 m, 880 s, 825 m, 756 m; ^1^H NMR (500 MHz, DMSO-*d*_6_, δ /ppm): 5.96 (s, 1H, H4), 7.45–8.23 (m, 10H, Ar-H); UV-Vis (λ_max_, nm): 360; Anal. Calculated for C_21_H_11_BrCl_2_HgN_4_O_4_: C 34.33, H 1.51, N 7.63, Found C 34.40, H 1.51, N 7.69%.

2-(5-(2-Chlorophenyl)-3-(4-methyl phenyl)-1H-pyrazol-1-yl)-3,5-dinitro phenyl) mercury(II) chloride (**2**)

Bright-yellow crystalline solid; Yield: 65%; M.p.: 138–140 °C; Molar conductance (Λ_m_, ohm^−1^ cm^−1^ mol^−1^): 31; R_f_ = 0.68 (ethyl acetate-n-hexane); FTIR (KBr) cm^−1^: 3063 w, 2974 w, 2913 w, 1600 s, 1465 s, 1438 s, 1392 s, 1323 m, 1273 s, 1107 s, 1064 s, 1030 m, 976, 821 m, 756 m; ^1^H NMR (500 MHz, DMSO-*d*_6_, δ /ppm): ^1^H NMR (500 MHz, DMSO-*d*_6_, δ/ppm): 2.20(s, 3H, CH_3_); 5.90 (s, 1H, H4), 7.50–8.20 (m, 10H, Ar–H); UV-Vis (λ_max_, nm): 350; Anal. Calculated for C_22_H_14_Cl_2_HgN_4_O_4_:C 32.98, H 1.76, N 6.99, Found: C 33.02, H 1.80, N 7.01%.

2-(5-(2-Chlorophenyl)-3-(4-methoxyphenyl)-1H-pyrazol-1-yl)-3,5-dinitrophenyl)mercury(II) chloride (**3**)

Light-yellow crystalline solid; Yield: 72%; M.p.: 180–182 °C; Molar conductance (Λ_m_, ohm^−1^ cm^−1^ mol^−1^): 39; R_f_ = 0.51 (ethyl acetate-n-hexane); FT-IR (KBr) cm^−1^: 3063 w, 2974 w, 2931 w, 2839 w, 1600 s, 1604 s, 1570 s,1508 s, 1462 s, 1427 s, 1327 s, 1257 s, 1222 m, 1180 s, 1111 s, 1030 s, 976 m, 825 m, 752 m, 678 m; ^1^H NMR (500 MHz, DMSO-*d*_6_, δ /ppm): 3.83(s, 6H, OCH_3_), 5.81 (s, 2H, H4 and H4’), 7.43–8.22 (m, 10H, Ar–H); UV-Vis (λ_max_, nm): 348; Anal. Calculated for C_22_H_14_Cl_2_HgN_4_O_5_:C 38.41, H 2.34, N 8.14, Found: C 38.47, H 2.39, N 8.21%.

#### 3.2.2. General Method for the Preparation of Aryl Tellurium Tribromides

(2-(3-(4-Substitutedphenyl)-5-(2-chlorophenyl)-1H-pyrazol-1-yl)-3,5-dinitrophenyl)tellurium tribromide **4**–**6**

A mixture of tellurium tetrabromide (1.78 g, 4.00 mmol) in 35 mL of dry dioxane and (4.00 mmol) aryl mercuric chlorides **1**, **2**, or **3**, respectively, in 30 mL of dry dioxane was refluxed with stirring for 6 h under an argon atmosphere. The resulting solution was filtered hot and on cooling deposited in a 2:1 complex of dioxane and mercuric chloride as white plates, which was filtered off. The filtrate was reduced by a rotary evaporator to give a brown precipitate. Recrystallization of the crude product from a mixture of chloroform and hexane (1:4) gave a yellow crystalline solid in 60–68% yields.

(2-(3-(4-Bromophenyl)-5-(2-chlorophenyl)-1H-pyrazol-1-yl)-3,5-dinitrophenyl)tellurium tribromide (**4**)

Light-yellowish-brown crystalline solid; Yield: 68%; M.p.: 212–214 °C; Molar conductance (Λ_m_, ohm^−1^ cm^−1^ mol^−1^): 27; R_f_ = 0.45 ethyl acetate-n-hexane); FTIR (KBr) cm^−1^: 3063 w, 1600 s, 1465 s, 1438 s, 1392 s, 1273 m, 1211 s, 1172 m, 1107 m, 1064 m, 1107 m, 1026, 976 m, 880 s, 821 m, 756 m; ^1^H NMR (500 MHz, DMSO-*d*_6_, δ/ppm): 5.96 (s, 1H, H4), 7.46–8.20 (m, 10H, Ar–H); UV-Vis (λ_max_, nm): 395; Anal. Calcd for C_21_H_11_Br_4_ClN_4_O_4_Te:C 29.13, H 1.28,N 6.47, Found: C 29.18,H 1.31, N 6.50%.

(2-(5-(2-Chlorophenyl)-3-(4-methylphenyl)-1H-pyrazol-1-yl)-3,5-dinitrophenyl)tellurium tribromide (**5**)

Light-yellowish-brown crystalline solid; Yield: 60%; M.p.: 159–161 °C; Molar conductance (Λ_m_, ohm^−1^ cm^−1^ mol^−1^): 29; Rf = 0.75 (ethyl acetate-n-hexane); FTIR (KBr) cm^−1^: 3075 m, 2925 w, 1592 m, 1463 m, 1439 m, 1396 m, 1325 s, 1277 m, 1212 m,1172 m, 1065 m, 1026 m, 978 s, 820 m, 756 s, 710 m, 685 m; ^1^H NMR (500 MHz, DMSO-*d*_6_, δ/ppm): 2.20(s, 3H, CH_3_); 5.92 (s, 1H, H4), 7.45–8.25 (m, 10H, Ar–H); UV-Vis (λ_max_, nm): 382; Anal. Calculated for C_22_H_14_Br_3_ClN_4_O_4_Te: C 32.98, H 1.76, N 6.99, Found: C 33.02, H 1.80, N 7.01%.

(2-(5-(2-Chlorophenyl)-3-(4-methoxyphenyl)-1H-pyrazol-1-yl)-3,5dinitrophenyl)tellurium tribromide (**6**)

Light-yellowish brown solid; Yield: 67%; M.p.: 195–197 °C; Molar conductance (Λ_m_, ohm^−1^ cm^−1^ mol^−1^): 26; R_f_ = 0.61 (ethyl acetate- n-hexane); FTIR (KBr) cm^−1^: 3063 w, 2904 w, 2833 w, 1604 s, 1570 s, 1512 m, 1465 m, 1427 s, 1327 s, 1264 m, 1226 m, 1194 m, 1114 m, 1003 s, 914 m, 823 m, 754 m; ^1^H NMR (500 MHz, DMSO-*d*_6_, δ /ppm): 3.86(s, 3H, OCH_3_), 5.84 (s, 1H, H4), 7.45–8.20 (m, 10H, Ar-H); UV-Vis (λ_max_, nm): 379; Anal. Calculated for C_22_H_14_Br_3_ClN_4_O_4_Te: C 32.34, H 1.73, N 6.86, Found: C 32.30, H 1.78, N 6.99%.

#### 3.2.3. General Method for the Preparation of Diaryl Tellurium Dibromides

Bis[(2-(3-(4-substitutedphenyl)-5-(2-chlorophenyl)-1H-pyrazol-1-yl)-3,5-dinitrophenyl)]tellurium dibromide **7**–**9**

A mixture of tellurium tetrabromide (0.89 g, 2.00 mmol) and aryl mercuric chloride **1**, **2**, or **3** (4.00 mmol) in 35 mL of dry dioxane was refluxed with stirring for 6 h under an argon gas atmosphere. The resulting solution was filtered hot and cooled to room temperature. On cooling, a 2:1 complex of dioxane and mercuric halides was separated as white plates and was filtered off immediately. Recrystallization of the product from a mixture of dichloromethane and hexane (1:4) gave an orange-brown to yellowish brown solid in 70–75% yield.

Bis[(2-(3-(4-bromophenyl)-5-(2-chlorophenyl)-1H-pyrazol-1-yl)-3,5-dinitrophenyl)]tellurium dibromide (**7**)

Light-orange-brown crystalline solid; Yield: 75%; M.p.: 209–211 °C; Molar conductance (Λ_m_, ohm^−1^ cm^−1^ mol^−1^): 32; R_f_ = 0.50 (ethyl acetate-n-hexane); FTIR (KBr) cm^−1^: 3059 w, 1604 s, 1570 s, 1516 s, 1465 m, 1438 m, 1338 s, 1311 m, 1273 m, 1211 s, 1180 m, 1157 m, 1041 s, 972 s, 860 s, 790 m, 752 m, 717 m, 690 m, 655 m, 578 m; ^1^H NMR (500 MHz, DMSO-*d*_6_, δ/ppm): 5.95 (s, 2H, H4 and H4′), 7.46–8.20 (m, 20H, Ar–H); UV-Vis (λ_max_, nm): 380; Anal. Calculated for C_42_H_22_Br_4_Cl_2_N_8_O_8_Te: C 39.26, H 1.73, N 8.72, Found: C 39.30, H 1.75, N 8.74%.

Bis[(2-(5-(2-Chlorophenyl)-3-(4-methylphenyl)-1H-pyrazol-1-yl)-3,5-dinitrophenyl)]tellurium dibromide (**8**)

Light-yellowish brown crystalline solid; Yield: 70%; M.p.: 150–152 °C; Molar conductance (Λ_m_, ohm^−1^ cm^−1^ mol^−1^): 25; R_f_ = 0.40 (ethyl acetate-n-hexane); FTIR (KBr) cm^−1^: 3063 m, 2924 w, 1604 s, 1593 m, 1512 m, 1469 m, 1442 m, 1315 s, 1273 m, 1215 m, 1037 m, 1014 m, 976 m, 752 m, 578 m; ^1^H NMR (500 MHz, DMSO-*d*_6_, δ/ppm): 2.06(s, 6H, 2CH_3_); 5.92 (s, 2H, H4 and H4′), 7.45–8.20 (m, 20H, Ar–H); UV-Vis (λ_max_, nm): 373; Anal. Calculated for C_44_H_28_Br_2_Cl_2_N_8_O_8_Te: C 45.75, H 2.44, N 9.70, Found: C 45.84, H 2.51, N 9.76%.

Bis[2-(5-(2-Chlorophenyl)-3-(4-methoxyphenyl)-1H-pyrazol-1-yl)-3,5dinitro phenyl)]tellurium dibromide (**9**)

Light-yellowish brown crystalline solid; Yield: 71%; M.p.: 191–193 °C; Molar conductance (Λ_m_, ohm^−1^ cm^−1^ mol^−1^): 30; Rf = 0.70 (ethyl acetate-n-hexane); FTIR (KBr) cm^−1^: 3063 w, 2931 w, 2850 w, 1606 s, 1571 s, 1520 m, 1460 m, 1430 m, 1330 s, 1261 m, 1227 m, 1180 m, 1034 m, 1014 m, 978 s, 830 m, 759 m, 682 m, 578 m; ^1^H NMR (500 MHz, DMSO-*d*_6_, δ/ppm): 3.86 (s, 6H, 2OCH_3_), 5.80 (s, 2H, 2H4), 7.45–8.20 (m, 20H, Ar–H); UV-Vis (λ_max_, nm): 371; Anal. Calculated for C_44_H_28_Br_2_Cl_2_N_8_O_10_Te: C 44.52, H 2.38, N 9.44, Found: C 44.60, H 2.41, N 9.45%.

#### 3.2.4. General Method for the Preparation of Diaryl Ditellurides

Bis[(2-(3-(4-substituted phenyl)-5-(2-chlorophenyl)-1H-pyrazol-1-yl)-3,5-dinitrophenyl)] ditelluride **10**–**12**

Aryl tellurium tribromide (3.00 mmol) was refluxed in ethanol (25 mL). An ethanolic solution of hydrazine hydrate was added drop by drop to the refluxing solution until the evolution of nitrogen ceased. The resulting solution was cooled to room temperature and poured into 100 mL of distilled water and extract with diethyl ether (4 × 30 mL). The etheric extracts were dried over an anhydrous calcium chloride. Evaporation of solvent afforded a dark red solid of compounds. The resulting precipitate was recrystallized by ethanol and gave a dark red solid in 61–68% yields.

Bis[(2-(3-(4-bromophenyl)-5-(2-chlorophenyl)-1H-pyrazol-1-yl)-3,5-dinitrophenyl)] ditelluride (**10**)

Dark red crystalline solid; Yield: 68%; M.p.: 100–102 °C; Molar conductance (Λ_m_, ohm^−1^ cm^−1^ mol^−1^): 9; R_f_ = 0.62 (ethyl acetate-n-hexane); FTIR (KBr) cm^−1^: 3072 w, 2928 w, 28431 w, 1600 s, 1570 s, 1520 m, 1465 m, 1431 m, 1331 s, 1266 m, 1226 m, 1181 m, 1033 m, 1010 m, 976 s, 829 m, 758 m, 684 m, 578 m; ^1^H NMR (500 MHz, DMSO-*d*_6_, δ /ppm): 5.96 (s, 2H, H4, and H4’), 7.47–8.23 (m, 20H, Ar–H); UV-Vis (λ_max_, nm): 500; Anal. Calculated for C_42_H_22_Br_2_Cl_2_N_8_O_8_Te_2_: C 40.27, H 1.77, N 8.95, Found: C 40.33, H 1.80, N 9.00%.

Bis[(2-(5-(2-Chlorophenyl)-3-(4-methylphenyl)-1H-pyrazol-1-yl)-3,5-dinitrophenyl)] ditelluride (**11**)

Dark red crystalline solid; Yield: 61%; M.p.: 91–93 °C; Molar conductance (Λ_m_, ohm^−1^ cm^−1^ mol^−1^): 11; R_f_ = 0.68 (ethyl acetate-n-hexane); FTIR (KBr) cm^−1^: 3063 w, 2924 w, 1631 s, 1569 m, 1539 m, 1481 m, 1455 m, 1399 m, 1335 s, 1269 s, 1211 m, 1134 m, 1068 m, 1045 m, 852 m, 826 m, 768 m, 628 w; ^1^H NMR (500 MHz, DMSO-*d*_6_, δ/ppm): 2.33(s, 6H, 2CH_3_); 5.91 (s, 2H, H4 and H4’), 7.42–8.20 (m, 20H, Ar–H); UV-Vis (λ_max_, nm): 491; Anal. Calculated for C_44_H_28_Cl_2_N_8_O_8_Te_2_: C 47.07, H 2.51, N 9.98, Found: C 47.10, H 2.50, N 10.10%.

Bis[(2-(5-(2-Chlorophenyl)-3-(4-methoxyphenyl)-1H-pyrazol-1-yl)-3,5dinitro phenyl)] ditelluride (**12**)

Dark red crystalline solid; Yield: 62%; M.p.: 99–101 °C; Molar conductance (Λ_m_, ohm^−1^ cm^−1^ mol^−1^): 10; R_f_ = 0.55 (ethyl acetate- n-hexane); IR (KBr) cm^−1^: 3063 w, 2974 w, 2931 w, 2839 w, 1604 m, 1570 s, 1512 m, 1485 m,1427 m, 1327 m, 1261 s, 1226 m, 1184 m, 1033 s, 1014 m, 976 m, 825 m, 756 m, 682 m, 578 m; ^1^H NMR (500 MHz, DMSO-*d*_6_, δ/ppm) 3.86 (s, 6H, 2OCH_3_), 5.80 (s, 2H, 2H4), 7.45–8.20 (m, 20H, Ar–H); UV-Vis (λ_max_, nm): 490; Anal. Calculated for C_44_H_28_Cl_2_N_8_O_10_Te_2_: C 45.70, H 2.35, N 9.68, Found: C 45.76, H 2.44, N 9.70%.

#### 3.2.5. General Method for the Preparation of Diaryl Tellurides

Bis[(2-(3-(4-substituted phenyl)-5-(2-chlorophenyl)-1H-pyrazol-1-yl)-3,5-dinitrophenyl)] telluride **13**–**15**

Diaryl tellurium dibromides (i.e., compounds **7**, **8**, or **9**) (2.00 mmol) was dissolved in 25 mL of ethanol and refluxed. A solution of hydrazine hydrate in ethanol was added drop wisely to the refluxed solution until nitrogen evaluation ceased. The resulting solution was poured into 500 mL of distilled ice water to afford a yellow solid. The crude product was twice recrystallized from a mixture of ethanol and dichloromethane to obtain a yellow or yellowish brown solid in 58–67% yields.

Bis[(2-(3-(4-bromophenyl)-5-(2-chlorophenyl)-1H-pyrazol-1-yl)-3,5-dinitrophenyl)] telluride (**13**)

Light-yellowish brown crystalline solid; Yield: 67%; M.p.: 88–90 °C; Molar conductance (Λ_m_, ohm^−1^ cm^−1^ mol^−1^): 13; R_f_ = 0.73 (ethyl acetate-n-hexane); FTIR (KBr) cm^−1^: 3063 w, 1609 s, 1597, 1465 m, 1438 m, 1323 s, 1273 m, 1311 m, 1272 m, 1211 s, 1181 m, 1157 m, 1041 s, 986 s, 860 s, 790 m, 756 m, 719 m, 667 m, 655 m, 578 m; ^1^H NMR (500 MHz, DMSO-*d_6_*, δ/ppm): 5.94 (s, 2H, H4 and H4′), 7.44–8.23 (m, 20H, Ar–H); UV-Vis (λ_max_, nm): 358; Anal. Calculated for C_21_H_11_BrCl_3_N_4_O_4_Te: C 34.43, H 1.51, N 7.65, Found: C 34.50, H 1.51, N7.71%.

Bis[(2-(5-(2-Chlorophenyl)-3-(4-methylphenyl)-1H-pyrazol-1-yl)-3,5-dinitrophenyl)] telluride (**14**)

Light-yellow crystalline solid; Yield: 58%; M.p.: 80–82 °C; Molar conductance (Λ_m_, ohm^−1^ cm^−1^ mol^−1^): 8; R_f_ = 0.65 (ethyl acetate-n-hexane); FTIR (KBr) cm^−1^: 3059 w, 1600 s, 1567 s, 1513 s, 1460 m, 1440 m, 1332 s, 1310 m, 1270 m, 1210 s, 1181 m, 1154 m, 1040 s, 970 s, 861 s, 788 m, 751 m, 719 m, 691 m, 657 m, 573 m; ^1^H NMR (500 MHz, DMSO-*d*_6_, δ /ppm): 2.06 (s, 6H, 2CH_3_); 5.90 (s, 2H, H4 and H4′), 7.46-8.20 (m, 20H, Ar–H); UV-Vis (λ max, nm): 345; Anal. Calculated for C_44_H_28_Cl_2_N_8_O_8_Te: C 53.10, H 2.84, N 11.26, Found: C 53.13, H 2.90, N 11.33%.

Bis[(2-(5-(2-Chlorophenyl)-3-(4-methoxyphenyl)-1H-pyrazol-1-yl)-3,5-dinitrophenyl)] telluride (**15**)

Light-yellow crystalline; Yield: 65%; M.p.: 87–85 °C; Molar conductance (Λ_m_, ohm^−1^ cm^−1^ mol^−1^): 11; R_f_ = 0.60 (ethyl acetate-n-hexane); FTIR (KBr) cm^−1^: 3063 w, 2974 w, 2931 w, 2839 w, 1604 m, 1570 s, 1512 s, 1465 m, 1427 m, 1327 s, 1261 m, 1226 m, 1180 m, 1111 m, 1033 m, 1014 m, 976 s, 825 m, 756 m, 682 m, 578 m, 505 m; ^1^H NMR (500 MHz, DMSO-*d*_6_, δ/ppm): 3.83 (s, 6H, 2OCH_3_), 5.81 (s, 2H, H4, and H4′), 7.43–8.22 (m, 20H, Ar–H); UV-Vis (λ_max_, nm): 341; Anal. Calculated for C_44_H_28_Cl_2_N_8_O_10_Te: C 51.45, H 2.75, N 10.91, Found: C 51.19, H 2.82, N 11.01%.

#### 3.2.6. General Method for the Preparation of Diaryl Tellurium Trichlorides

(2-(3-(4-substituedphenyl)-5-(2-chlorophenyl)-*1H*-pyrazol-1-yl)-3,5-dinitrophenyl)tellurium trichloride **16**–**21**

Thionyl chloride (0.12 g, 1.00 mmol) in 15 mL of ethanol was added drop wisely to an ethanolic solution of diaryl ditellurides compounds (i.e., compounds **10**, **11** or **12**) (1.00 mmol) with stirring at room temperature for 30 minutes. A yellow precipitate was formed immediately. Recrystallization by ethanol gave an yellow solid of compounds **16**–**18**.

(2-(3-(4-Bromophenyl)-5-(2-chlorophenyl)-1H-pyrazol-1-yl)-3,5-dinitrophenyl)tellurium trichloride (**16**)

Yellow brown crystalline solid; 82 Yield: %; M.p.: 168–170 °C; Molar conductance (Λ_m_, ohm^−1^ cm^−1^ mol^−1^): 33; R_f_ (ethyl acetate- n-hexane) = 0.81; FT-IR (KBr) cm^−1^: 3059 w, 1604 s, 1570 s, 1465 s, 1438 s, 1138 s, 1311 s, 1273 s, 1211 s, 1180 s, 1157 s, 1041 m, 972 s, 850 m, 790 m, 717 m, 748 s, 717 m, 690 m, 659 m; ^1^H NMR (500 MHz, DMSO-*d*_6_, δ/ppm): 5.95 (s, 2H, H4 and H4′), 7.45–8.20 (m, 20H, Ar-H); UV-Vis (λ_max_, nm): 400; Anal. Calculated for C_21_H_11_BrCl_4_N_4_O_4_Te: C 51.45, H 2.75, N 10.91, Found: C 51.19, H 2.82, N 11.01%.

(2-(3-(4-Methylphenyl)-5-(2-chlorophenyl)-1H-pyrazol-1-yl)-3,5-dinitrophenyl)tellurium trichloride (**17**)

Yellow crystalline solid; 76 Yield: %; M.p.: 160–162 °C; Molar conductance (Λ_m_, ohm^−1^ cm^−1^ mol^−1^): 30; R_f_ (ethyl acetate- n-hexane) = 0.75; FTIR (KBr) cm^−1^: FT-IR (KBr) cm^−1^: 3065 m, 2930 w, 1631 m, 1590 m, 1463 m, 1433 m, 1390 m, 1327 s, 1275 m, 1216 m,1170 m, 1066 m, 1022 m, 971 s, 821 m, 756 s, 710 m, 687 m; ^1^H NMR (500 MHz, DMSO-*d*_6_, δ/ppm): 2.20 (s, 6H, 2CH_3_), 5.90 (s, 2H, H4 and H4′), 7.45–8.20 (m, 20H, Ar–H); UV-Vis (λ_max_, nm): 375; Anal. Calculated for C_22_H_14_Cl_4_N_4_O_4_Te: C 38.64, H 2.06, N 8.19, Found: C 38.70, H 2.00, N 8.24%.

(2-(3-(4-Methoxyphenyl)-5-(2-chlorophenyl)-1H-pyrazol-1-yl)-3,5-dinitrophenyl)tellurium trichloride (**18**)

Yellow crystalline solid; 79 Yield: %; M.p.: 155–156 °C; Molar conductance (Λ_m_, ohm^−1^ cm^−1^ mol^−1^): 24; R_f_ (ethyl acetate- n-hexane) = 0.70; FTIR (KBr) cm^−1^: 3060 w, 2911 w, 2835 w, 1600 s, 1568 s, 1511 m, 1462 m, 1425 s, 1324 s, 1265 m, 1230 m, 1195m, 1112 m, 1005 s, 914 m, 825 m, 759 m; ^1^H NMR (500 MHz, DMSO-d_6_, δ/ppm): ^1^H NMR (500 MHz, DMSO-*d*_6_, δ /ppm): 3.86 (s, 3H, OCH_3_), 5.82 (s, 1H, H4), 7.45–8.22 (m, 10H, Ar–H); UV-Vis (λ_max_, nm): 370; Anal. Calculated for C_22_H_14_Cl_4_N_4_O_4_Te: C 51.45, H 2.75, N 10.91, Found: C 51.19, H 2.82, N 11.01%.

#### 3.2.7. General Method for the Preparation of Diaryl Tellurium Triiodides

(2-(3-(4-Substitutedphenyl)-5-(2-chlorophenyl)-1*H*-pyrazol-1-yl)-3,5-dinitrophenyl)tellurium triiodide **19**–**21**

A solution of iodine (0.10 g, 0.78 mmol) in 10 mL of ethanol added to a solution of diaryl ditelluride compounds **10**, **11**, or **12** (0.78 mmol) in 20 mL ethanol with stirring at room temperature for 30 min gave a brown solid of compounds **19**, **20**, and **21**, respectively.

(2-(3-(4-Bromophenyl)-5-(2-chlorophenyl)-1*H*-pyrazol-1-yl)-3,5-dinitrophenyl)tellurium triiodide (**19**)

Yellowish brown crystalline solid; 73 Yield: %; M.p.: 147–159 °C; Molar conductance (Λ_m_, ohm^−1^ cm^−1^ mol^−1^): 30; R_f_ (ethyl acetate-n-hexane) = 0.45; FTIR (KBr) cm^−1^: FTIR (KBr) cm^−1^: 3055 w, 1591 s, 1460 s, 1439 s, 1391 s, 1268 m, 1214 s, 1177 m, 1100 m, 1065 m, 1105 m, 1025, 971 m, 882 s, 820 m, 756 m; ^1^H NMR (500 MHz, DMSO-*d*_6_, δ/ppm): 5.96 (s, 2H, H4, and H4′), 7.45–8.22 (m, 20H, Ar–H); UV-Vis (λ_max_, nm): 390; Anal. Calculated for C_21_H_11_I_3_BrClN_4_O_4_Te: C 25.05, H 1.10, N 5.56, Found: C 25.09, H 1.17, N 5.63.01%.

(2-(3-(4-Methylphenyl)-5-(2-chlorophenyl)-1*H*-pyrazol-1-yl)-3,5-dinitrophenyl)tellurium triiodide (**20**)

Yellow crystalline solid; 76 Yield: %; M.p.: 122–120 °C; Molar conductance (Λ_m_, ohm^−1^ cm^−1^ mol^−1^): 29; R_f_ (ethyl acetate-n-hexane) = 0.61; FTIR (KBr) cm^−1^: 3063 m, 2924 w, 1597 m, 1465 m, 1438 m, 1392 m, 1323 s, 1273 m, 1211 m,1172 m, 1064 m, 1026 m, 978 s, 821 m, 756 s, 709 m, 687 m; ^1^H NMR (500 MHz, DMSO-*d*_6_, δ/ppm): 2.22 (s, 3H, CH_3_); 5.90 (s, 1H, H4), 7.46–8.20 (m, 10H, Ar-H); UV-Vis (λ_max_, nm): 395; Anal. Calculated for C_22_H_14_I_3_ClN_4_O_4_Te: C 27.58, H 1.50, N 5.95, Found: C 28.10, H 1.51, N 6.01%.

(2-(3-(4-Methoxyphenyl)-5-(2-chlorophenyl)-1*H*-pyrazol-1-yl)-3,5-dinitrophenyl)tellurium triiodide (**21**)

Yellowish brown crystalline solid; 70 Yield: %; M.p.: 113–105 °C; Molar conductance (Λ_m_, ohm^−1^ cm^−1^ mol^−1^): 22; R_f_(ethyl acetate-n-hexane) = 65; FTIR (KBr) cm^−1^: 3059 w, 2905 w, 2835 w, 1598 s, 1570 s, 1515 m, 1465 m, 1428 s, 1327 s, 1265 m, 1227 m, 1192 m, 1117 m, 1013 s, 915 m, 820 m, 755 m; ^1^H NMR (500 MHz, DMSO-*d*_6_, δ/ppm): 3.8 (s, 6H, 2OCH_3_), 5.81 (s, 2H, H4, and H4′), 7.43–8.22 (m, 20H, Ar–H); UV-Vis (λ_max_, nm): 391; Anal. Calculated for C_22_H_14_I_3_ClN_4_O_4_Te: C 27.58, H 1.47, N 5.85, Found: C 27.62, H 1.41, N 6.94%.

#### 3.2.8. General Method for the Preparation of Diaryl Tellurium Dichlorides

Bis[(2-(3-(4-substitutedphenyl)-5-(2-chlorophenyl)-1H-pyrazol-1-yl)-3,5-dinitrophenyl)]tellurium dichlorides **22**–**24**

Thionyl chloride (0.12 g, 1.00 mmol) in 15 mL of ethanol was added drop wise to an ethanolic solution of diaryl tellurides compounds (i.e., compounds **13**, **14**, or **15**) (1.00 mmol) with stirring at room temperature for 30 minutes. A yellow precipitate was formed immediately. Recrystallization by ethanol gave a yellow solid of compounds **22**–**24**.

Bis[(2-(3-(4-bromophenyl)-5-(2-chlorophenyl)-1H-pyrazol-1-yl)-3,5-dinitrophenyl)]tellurium dichloride (**22**)

Yellowish orange crystalline solid; Yield: 83%; M.p.: 187–180 °C; Molar conductance (Λ_m_, ohm^−1^ cm^−1^ mol^−1^): 20; R_f_ = 0.60 (ethyl acetate-n-hexane); FTIR (KBr) cm^−1^: 3063 w, 1660 s, 1609 s, 1597, 1465 m, 1438 m, 1323 s, 1273 m, 1311 m, 1272 m, 1211 s, 1181 m, 1157 m, 1041 s, 986 s, 860 s, 790 m, 756 m, 719 m, 667 m, 655 m, 578 m; ^1^H NMR (500 MHz, DMSO-*d*_6_, δ/ppm): 5.95 (s, 2H, H4, and H4′), 7.46–8.20 (m, 20H, Ar–H); UV-Vis (λ_max_, nm): 385; Anal. Calculated for C_42_H_22_Br_2_Cl_4_N_8_O_8_Te: C 42.18, H 1.85, N 9.37 Found: C 42.20, H 1.87, N 9.37%.

Bis[(2-(3-(4-methylphenyl)-5-(2-chlorophenyl)-1H-pyrazol-1-yl)-3,5-dinitrophenyl)]tellurium dichloride (**23**)

Yellow crystalline solid; Yield: 73%; M.p.: 104–106 °C; Molar conductance (Λ_m_, ohm^−1^ cm^−1^ mol^−1^): 21; R_f_ (ethyl acetate-n-hexane) = 0.58; FTIR (KBr) cm^−1^: 3063 w, 2927, 1600 s, 1570 s, 1485 s, 1438 m, 1323 s, 1273 m, 1211 m, 1172 m, 1064 s, 1007 m, 976 s, 820 m, 756 m, 709 m, 687 m, 582 m, 536 m; ^1^H NMR (500 MHz, DMSO-*d*_6_, δ/ppm): 2.06 (s, 6H, 2CH_3_); 5.92 (s, 2H, H4, and H4′), 7.45–8.20 (m, 20H, Ar–H); UV-Vis (λ_max_, nm): 385; Anal. Calculated for C_44_H_28_Cl_4_N_8_O_8_Te: C 49.57, H 2.65, N 10.51, Found: C 49.61, H 2.69, N 10.58%.

Bis[(2-(3-(4-methoxyphenyl)-5-(2-chlorophenyl)-1H-pyrazol-1-yl)-3,5-dinitrophenyl)]tellurium dichloride (**24**)

Light-yellowish crystalline solid; Yield: 78%; M.p.: 181–183 °C; Molar conductance (Λ_m_, ohm^−1^ cm^−1^ mol^−1^): 26; Rf (ethyl acetate-n-hexane) = 0.60; FTIR (KBr) cm^−1^: 3063 w, 2928 w, 2843 w, 1604 s, 1570 s, 1521 m, 1465 m, 1431 m, 1330 s, 1261 m, 1226 m, 1180 m, 1033 m, 1014 m, 976 s, 829 m, 758 m, 682 m, 578 m; ^1^H NMR (500 MHz, DMSO-*d*_6_, δ /ppm): 3.86 (s, 6H, 2OCH_3_), 5.80 (s, 2H, 2H4), 7.45–8.20 (m, 20H, Ar–H); UV-Vis (λ_max_, nm): 385; Anal. Calculated for C_44_H_28_Cl_4_N_8_O_10_Te: C 48.12, H 2.57, N 10.20, Found: C 48.18, H 2.62, N 10.20%.

#### 3.2.9. General Method for the Preparation of Diaryl Tellurium Diiodides

Bis[(2-(3-(4-substitutedphenyl)-5-(2-chlorophenyl)-1H-pyrazol-1-yl)-3,5-dinitrophenyl)]tellurium diiodides **25**–**27**

A solution of iodine (0.10 g, 0.78 mmol) in 10 mL of ethanol added to a solution of diaryl tellurides compounds **13**, **14**, or **15** (0.78 mmol) in 20 mL ethanol with stirring at room temperature for 30 min gave a brown solid of compounds **25**, **26**, and **27**, respectively.

Bis[(2-(3-(4-bromophenyl)-5-(2-chlorophenyl)-1H-pyrazol-1-yl)-3,5-dinitrophenyl)]tellurium diiodides (**25**)

Yellowish brown crystalline solid; Yield: 73%; M.p.: 113–115 °C; Molar conductance (Λ_m_, ohm^−1^ cm^−1^ mol^−1^): 27; R_f_ (ethyl acetate-n-hexane) = 0.66; FTIR (KBr) cm^−1^: 3059 w, 1604 s, 1523 m, 1469 m, 1438 m, 1334 s, 1311 m, 1269 m, 1211 m, 1010 m, 972 m, 860 m, 748 m, 721 m, 686 m, 655; ^1^H NMR (500 MHz, DMSO-*d*_6_, δ/ppm): 5.95 (s, 2H, H4 and H4′), 7.46–8.20 (m, 20H, Ar–H); UV-Vis (λ_max_, nm): 390; Anal. Calculated for C_42_H_22_I_2_Br_2_Cl_2_N_8_O_8_Te: C 27.52, H 1.68, N 5.84 Found: C 27.60, H 1.71, N 5.91%.

Bis[(2-(3-(4-methylphenyl)-5-(2-chlorophenyl)-1H-pyrazol-1-yl)-3,5-dinitrophenyl)]tellurium diiodides (**26**)

Yellowish brown crystalline solid; Yield: 77%; M.p.: 80–82 °C; Molar conductance (Λ_m_, ohm^−1^ cm^−1^ mol^−1^): 21; R_f_(ethyl acetate-n-hexane) = 0.66; FTIR (KBr) cm^−1^: 3059 m, 2920 w, 1604 s, 1523 m, 1462 m, 1438 m, 1334, 1311 s, 1269 m, 1211 s, 1037 m, 1010 m, 972 m, 860, 748 s, 721 m, 585 m, 655 m, 578 m, 528 m, 447 m; ^1^H NMR (500 MHz, DMSO-*d*_6_, δ /ppm): 2.06 (s, 6H, 2CH_3_); 5.92 (s, 2H, H4, and H4′), 7.45–8.20 (m, 20H, Ar–H); UV-Vis (λ_max_, nm): 395; Anal. Calculated for C_44_H_28_I_2_Cl_2_N_8_O_8_Te: C 36.59, H 1.61, N 8.13, Found: C 37.04, H 1.66, N 8.15 %.

Bis[(2-(3-(4-methoxyphenyl)-5-(2-chlorophenyl)-1H-pyrazol-1-yl)-3,5-dinitrophenyl)]tellurium diiodides (**27**)

Yellowish brown crystalline solid; Yield: 66%; M.p.: 93–95 °C; Molar conductance (Λ_m_, ohm^−1^ cm^−1^ mol^−1^): 19; Rf (ethyl acetate-n-hexane) = 0.60; FTIR (KBr) cm^−1^: 3056 w, 2928 w, 2840 w, 1600 s, 1572 s, 1521 m, 1465 m, 1430 m, 1334 s, 1260 m, 1226 m, 1180 m, 1032 m, 1016 m, 976 s, 829 m, 758 m, 682 m, 577 m; ^1^H NMR (500 MHz, DMSO-*d*_6_, δ/ppm): 3.83 (s, 6H, 2OCH_3_), 5.82 (s, 2H, H4, and H4′), 7.46–8.22 (m, 20H, Ar–H); UV-Vis (λ_max_, nm): 390; Anal. Calculated for C_44_H_28_I_4_Cl_2_N_8_O_10_Te: C 42.31, H 2.26, N 8.97, Found: C 42.39, H 2.31, N 9.02%.

### 3.3. Determination of Antimicrobial Activity

Compounds **1**–**27** were assayed against four different microorganisms: Gram-positive bacteria *Staphylococcus aureus* ATCC25923 and *Candida albicans* ATCC2091, and Gram-negative bacteria *Escherichia coli* ATCC25922 and *Pseudomonas aeruginosa* ATCC9027 by using the disk diffusion technique. Amoxicillin (10 µg/disc) was used as the standard drug. The agar well diffusion method was applied for the determination of the inhibition zone and minimum inhibitory concentration (MIC). The procedure used in this study was according to a previous method [4].

## 4. Conclusions

This work shows an efficient method for the synthesis of a series of new tellurated compounds derived from 1, 3, 5-trisubstituted-pyrazole derivatives for the first time. The compounds that possess pharmacophores such as bromo-, methoxy- and methyl-substituents with lipophilic properties appeared to have the greatest antimicrobial activity. The diaryl tellurides **13**, **14**, and **15** were highly active against the organisms employed among all synthesized compounds. Compounds **13**–**15** possess more potent activity against the bacteria than control (amoxicillin), which makes them promising drugs in the future.

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
