# Peer review of "Synthesis and Antimicrobial Evaluation of Some New Organic Tellurium Compounds Based on Pyrazole Derivatives"

_molecules, 2020, doi:10.3390/molecules25153439_

Round 1
Reviewer 1 Report
This Asmaa B. Sabti , Adil A. Al-Fregi * , Majeed Y. Yousif paper describes an study about symtheis and antimicrobial activity of various Te compounds with different pyrazole ligands. A similar class of compounds have been reported in previous papers (Phosphorus sulfur and silicon and the related elements 194, 33-38, 2019).
The synthesis of the proligands from chalcones and the tellurium compounds by transmetallation have been studied. The use of different characterization techniques is performed. Antimicrobial studies and the effect of the nature of the susbstituents in the ligands is explained.
I recommended this manuscript for publication after major revision.
1) Concerning to typing errors:
Line 32, size letter must be revised.
Scheme 1: different types of letter in the graphic.
Line 56: verb is missing: are generated instead generated.
Line 58: similar manner instead similar mannar
Scheme 3 and 4: I2 not I2.
Line 77: reaction
Line 91: pyrazole
Line 147: which facilitates not which it facilitates
Line 493: procedures
2) On line 76 is written “result in single diastereomer” This fact is concluded by TLC analysis? It must be corroborated by NMR studies. Do you know why these reactions are diastereoselective?
3) Compounds 10- 12, as I can deduct from the pictures, present a Te-Te bond. But in this case what is the oxidation state of Te?
4) All NMR spectra have been made in DMSO-d6, this solvent can affect the nuclearity of the obtained complexes. It could be interesting doing some of the spectra in another solvent.
5) These complexes show a potent activity against both Gram positive and Gram negative bacteria in different degrees. Have you considered the effect of tellurium toxicity without ligand? The authors reclaims the better effect to the more liphophilicity caused by the substituents of the ligands, in this case have you done some assay without metal , only with the proligands?
Author Response
Please see the attachment file
with all respect for you

Reviewer 2 Report
Opinion on the paper: „Synthesis and Antimicrobial Evaluation of Some New Organic Tellurium Compounds Based on Pyrazole Derivatives”
Authors have synthesized a series of new organic tellurium compounds possessing antimicrobial activity. The compounds were characterized by means of NMR, FT-IR and UV-VIS spectroscopies, melting point measurements and molar conductance. In my opinion, the compounds have been properly characterized and their structures have been proven. Because of poor English usage, the paper may be accepted for publication after a language correction. Below is a list of language errors, but it is not complete. The Authors should use the help of a professional proofreader.
Page 1 line 42: it should be “so, in the present work” instead of “so the present work”
Page 2, Line 50: remove “}”
Line 58: it should be “ in a similar manner “ instead of “A similar mannar”
Line 60: it should be “ resulted in ” instead of “obtained”
Line 64: “compounds” should be removed
Page 3, line 67: it should be “ The reaction of diaryl ditellurides 10-12 with an excess ” instead of “Reaction diaryl ditellurides 10-12 with excess”
Page 4, line 77: it should be “ reaction ” instead of “ reacton”
Line 91: it should be “ the spectra of pyrazole” instead of “ The spectra of pyrzole”
Line 94: it should be “ characterized ” instead of “characterizied”
Line 101: it should be “ appear as ” instead of “appearas”
Line 102: it should be “ the spectra show ” instead of “show”
Line 105: remove “in well agreement”
Line 107: it should be “ reveal ” instead of “appear”
Line 108: it should be “ reveal ” instead of “appear”
Line 109: it should be “ which can ” instead of “can”
Line 110-112: the sentence is incomprehensible
Line 115: it should be “ increasing ” instead of “leads to increase”
Line 115 it should be “ resonance ” instead of “resonane”
Line 116: remove “can be”
Page 5, line 121: it should be “ of ” instead of “is due to”
Line 122: remove “can be”
Line 126: it should be “investigation ” instead of “invstigation”
Line 127: it should be “ causes ” instead of “cause”
Line 128: it should be “ absorption ” instead of “absorotion”
Line 131-133: it should be “ It was proven that the molar conductance of organic mercuric compounds 1-3, aryl tellurium tribromides 4-6 and 16-21 and the dibromides 7-9 and 22-27 behave as for 1:1 electrolyte which are in good agreement with the previous works in DMSO solutions [9,10,17,18].” instead of “The molar conductance of organic mercuric compounds 1-3, aryl tellurium tribromides 4-6 and 16-21 and the dibromides 7-9 and 22-27 were proved that these behave as 1:1 electrolyte which are in agree well with previous works in DMSO solution [9,10,17,18].”
Page 6,line 145: it should be “ active ” instead of “activity”
Page 13, line 486: it should be “ possess ” instead of “is”
Author Response

(The authors gave the same response as above.)

Reviewer 3 Report
The authors describe the reactivity of a family of pyrazole-containing organomercury chloride compounds towards TeBr4, yielding mono or disubstituted Te(IV) compounds. Reduction with hydrazine yields either diaryl tellurides or diaryl ditellurides. Oxidation of these reduced compounds with SOCl2 or I2 yields chloride or iodide Te(IV) compounds. The compounds are investigated with respect to antimicrobial activity.
The antimicrobial activity part is outside my area of expertise, so I will not comment on it.
I have three major points which need to be addressed before I can recommend publication:
- The authors claim that to their knowledge “there are no reported studies related to synthesis of organotellurium compounds derived from pyrazoles” However I was able to find the following reports containing such compounds. These references should be added to the manuscript and the claim adjusted to reflect them.
K. Bhasin, S. Pundir, S. Neogy, D. Mehta, S. K. Mehta, 2018, Phosphorus, Sulfur, and Silicon and the Related Elements, 193:5, 273-279, DOI: 10.1080/10426507.2017.1399127
Chandrasekhar, A. Kumar, M. D. Pandey, R. K. Metre, Polyhedron, 2013, 52, 1362-1368 https://doi.org/10.1016/j.poly.2012.06.005. - In the keeping with Molecules’ guidelines for including supplementary material https://www.mdpi.com/journal/molecules/instructions#suppmaterials Please include copies of the NMR, IR, and UV-vis spectra for all new compounds.
- The presentation of the schemes should be improved. In Scheme 1 the hydrazine is shown as an ArHgCl2 compound but should be ArHgCl instead. I recommend labelling with an index the two reactants which yield products 1-3. This will avoid having to refer to them by their IUPAC names, which are long and complicated. The structures represented in Scheme 2 have distorted bond angles and would benefit from some tidying. Care should be taken to ensure that, between schemes, the size of the aryl rings remains consistent.
One minor point: I am curious about the choice of starting materials to synthesize 1-3? Why did the authors choose to have an ortho-Cl group on the chalcone, or the two NO2 groups on the hydrazine?
Author Response
Please sir
see the attachment file
with my respect for you

Round 2
Reviewer 3 Report
The manuscript has improved significantly, and my recommendations have been addressed, except for my request to show spectra as supplementary material. The authors' excuse that there is a large number of figures does not persuade me. Including spectra for new compounds in a file separate from the manuscript is standard in synthetic chemistry. See this recent paper in Molecules: https://doi.org/10.3390/molecules25112571
I do not recommend publication of the manuscript in its current form.
